# A case report of multiple primary prostate tumors with differential drug sensitivity

Scott Wilkinson [1,10], Stephanie A. Harmon[2,3,10], Nicholas T. Terrigino[1], Fatima Karzai[4], Peter A. Pinto[5], Ravi A. Madan[4], David J. VanderWeele[1,8], Ross Lake[1], Rayann Atway [1], John R. Bright[1], Nicole V. Carrabba [1], Shana Y. Trostel[1], Rosina T. Lis[1], Guinevere Chun[4], James L. Gulley [4], Maria J. Merino[6], Peter L. Choyke[2], Huihui Ye[7,9], William L. Dahut[4], Baris Turkbey[2,11] & Adam G. Sowalsky [1,11✉]

Localized prostate cancers are genetically variable and frequently multifocal, comprising spatially distinct regions with multiple independently-evolving clones. To date there is no understanding of whether this variability can influence management decisions for patients with prostate tumors. Here, we present a single case from a clinical trial of neoadjuvant intense androgen deprivation therapy. A patient was diagnosed with a large semi-contiguous tumor by imaging, histologically composed of a large Gleason score 9 tumor with an adjacent Gleason score 7 nodule. DNA sequencing demonstrates these are two independent tumors, as only the Gleason 9 tumor harbors single-copy losses of *PTEN* and *TP53*. The *PTEN/TP53*-deficient tumor demonstrates treatment resistance, selecting for subclones with mutations to the remaining copies of *PTEN* and *TP53*, while the Gleason 7 *PTEN*-intact tumor is almost entirely ablated. These findings indicate that spatiogenetic variability is a major confounder for personalized treatment of patients with prostate cancer.

[1] Laboratory of Genitourinary Cancer Pathogenesis, National Cancer Institute, NIH, 37 Convent Drive, Bethesda, MD 20892, USA. [2] Molecular Imaging Program, National Cancer Institute, NIH, 10 Center Drive, Bethesda, MD 20892, USA. [3] Clinical Research Directorate, Frederick National Laboratory for Cancer Research sponsored by the National Cancer Institute, 8560 Progress Drive, Frederick, MD 21701, USA. [4] Genitourinary Malignancies Branch, National Cancer Institute, NIH, 10 Center Drive, Bethesda, MD 20892, USA. [5] Urologic Oncology Branch, National Cancer Institute, NIH, 10 Center Drive, Bethesda, MD 20892, USA. [6] Laboratory of Pathology, National Cancer Institute, NIH, 10 Center Drive, Bethesda, MD 20892, USA. [7] Department of Pathology, Beth Israel Deaconess Medical Center, 330 Brookline Avenue, Boston, MA 02215, USA. [8] Present address: Department of Medicine, Feinberg School of Medicine, 420 E. Superior Street, Chicago, IL 60611, USA. [9] Present address: Department of Pathology and Department of Urology, University of California Los Angeles, 10833 Le Conte Avenue, Los Angeles, CA 90095, USA. [10] These authors contributed equally: Scott Wilkinson, Stephanie A. Harmon. [11] These authors jointly supervised this work: Baris Turkbey, Adam G. Sowalsky. ✉email: adam.sowalsky@nih.gov

Localized prostate cancers are uniquely genetically variable and frequently multifocal. Retrospective DNA sequencing studies have demonstrated that 80–90% of prostate cancers comprise multiple spatially separate (distinct) regions, with occasional development of two independent clones within the same tumor[1,2]. This multifocality is a well-recognized problem for diagnosis, and to date there has been no understanding of whether this variability can influence personalized management decisions. Conventional management of high-risk localized prostate cancer includes radical prostatectomy or external beam radiation therapy with androgen deprivation therapy (ADT)[3]. The target of ADT is the transcription factor androgen receptor (AR), which mediates terminal differentiation of luminal epithelium in benign prostatic tissue, but drives cell proliferation in prostatic adenocarcinoma. Observations that AR-targeting agents (such as abiraterone and enzalutamide) improve overall and metastasis-free survival in patients with hormone-sensitive and nonmetastatic disease, respectively, suggests that earlier intense inhibition of AR may further improve outcomes[4–6]. To this end, the effect of introducing intense androgen deprivation therapies in the neoadjuvant setting has been explored in a series of clinical trials at several cancer centers[7]. Although overall survival benefits associated with neoadjuvant intense ADT are yet unknown, complete pathologic responses were reported in ~10% of patients, and minimal residual disease (MRD) has been observed in ~30% of patients[8]. Surgical specimens from ~90% of patients show treatment effect, with substantial residual disease in ~20% of cases, which are likely due to driver mutations present at baseline, such as mutations to *AR* itself[9], or selection for subclones with losses to tumor suppressors, including *PTEN* and *TP53*[10]. Despite evidence demonstrating repeated sampling of multifocal prostate cancer gives rise to discordant scores on prognostic tissue tests, the treatment outcomes of independent prostate cancer foci in a patient with a polyclonal or polytumor phenotype remain unknown[11].

Multiparametric magnetic resonance imaging (mpMRI) has become an established imaging technique for diagnosis and staging of localized prostate cancer, especially in detection and sampling of clinically significant disease while decreasing detection of indolent, low-risk tumors[12–14]. In our study of neoadjuvant ADT plus enzalutamide, we hypothesized that mpMRI prior to surgery would assess radiographic tumor response compared with baseline imaging, while also providing biopsy targets for molecular and histologic examination of pre-treatment foci. Given the high-risk population for whom neoadjuvant intense ADT is suggested to benefit, patients on our study with larger lesions on mpMRI undergo saturation biopsy of imaging targets, to more accurately assess tumor heterogeneity[15,16].

In this report, we describe a patient who presented with high-risk prostate cancer that exhibited differential intratumoral response to a clinical trial of neoadjuvant intense ADT. Histologic and genomic analyses reveal the presence of a polytumor, in which one nodule was mostly sensitive to treatment and a separate, clonally independent nodule was resistant. While comprehensive molecular profiling of multiple specimens from a single patient is uncommon for many biomarker-driven therapies, we propose that success of genomically driven trials and treatment strategies may require complete assessment of larger tumors prior to application of precision therapies.

## Results

### Identification of two primary prostate tumors at baseline.
In our phase 2 clinical study of neoadjuvant ADT and enzalutamide, patients with localized intermediate to high-risk prostate cancer undergo mpMRI, which is used for guiding the acquisition of biopsy tissue. Patients then receive intense hormonal therapy (goserelin plus enzalutamide) for 24 weeks, followed by a second mpMRI and radical prostatectomy (RP). The final results for this trial have yet not been reported.

In a single case from this trial, a 66-year-old patient's pre-treatment mpMRI (see Supplementary Table 1) identified a semi-contiguous tumor that extended from the right apical–mid peripheral zone to the left distal apical peripheral zone, although imaging alone did not provide sufficient evidence of two independent tumors during the initial mpMRI interpretation (Fig. 1a). Due to the extension of tumor across the midline with differential T2-weighted and *b*2000 MRI characteristics in the distal apical portion of the prostate (Supplementary Fig. 1a), the left- and right-sided tumor extent were separately sampled by MR/ultrasound (US)-fusion transrectal biopsy (Fig. 1b; Supplementary Fig. 1b). These biopsies showed markedly differing histologies (Fig. 1c). The left-sided tumor (B1) was Grade Group (GG) 3 (Gleason score $4 + 3 = 7$) with intermediately to well-differentiated adenocarcinoma. In contrast, the right-sided tumor (B2/B3) was GG5 (Gleason score $4 + 5 = 9$) poorly differentiated carcinoma with a component of intraductal carcinoma (IDC-P) (Supplementary Fig. 1c) and perineural invasion.

The TMPRSS2:ERG fusion, which occurs in 40–50% of prostate cancer patients and results in ERG expression, is an early genomic event during tumor development[17,18]. *PTEN* loss, represented by significant reduction or complete loss of PTEN expression, is another common alteration in prostate cancer pathogenesis and tends to co-occur after an ETS family fusion[18,19]. We therefore performed anti-ERG and anti-PTEN immunohistochemistry on serial sections of biopsy tissue (Fig. 1c; Supplementary Fig. 1c). The left-sided tumor was ERG-negative with reduced PTEN in a subset of tumor cells (B1), while the right-sided tumor stained positively for ERG and had homogenous reduction in PTEN expression in both the IDC-P (B2) and invasive (B3) components.

To determine if the genetic basis of reduced PTEN expression in the left- and right-sided tumors was due to the same genomic event, we performed laser capture microdissection (LCM) of tumor foci and adjacent normal tissue, and then performed whole-exome sequencing (WES) to assess somatic copy number alterations (SCNAs) and point mutations. As shown in Supplementary Fig. 2a and Supplementary Table 2, both foci from the right-sided tumor (B2 and B3) showed an arm-level deletion of chromosome 10q encompassing *PTEN* and a focal deletion in chromosome 17p encompassing *TP53*. However, the left-sided tumor (B1) showed no genomic hits to *PTEN* or *TP53* (Supplementary Fig. 2a), suggesting a noncoding or epigenetic driver of reduced PTEN expression. In addition to these mutually exclusive SCNAs, the B2 and B3 foci harbored shared losses of chromosome 16 not found in the B1 focus, and B1 carried a gain of chromosome 7 not detected in B2 or B3 (Supplementary Fig. 2a).

Examination of somatic point mutations resolved by WES revealed that the vast majority were harbored by only a single tumor focus, although nine point mutations were shared by the B2 and B3 foci (Supplementary Table 2). No mutations were shared by all three foci. Given the relatively shallow coverage from WES, we verified that these mutations were also definitive markers distinguishing the B1 and B2/B3 tumors by performing additional focused sequencing to an average depth of ~50,000×. We observed 29 point mutations that demonstrated mutual exclusivity between the left- and right-sided tumor foci (Supplementary Table 2), confirming the clonal independence of these two tumors.

### Identification of independent tumors after treatment.
Following baseline imaging and biopsy, the patient received 6 months of neoadjuvant intense ADT with goserelin and enzalutamide per

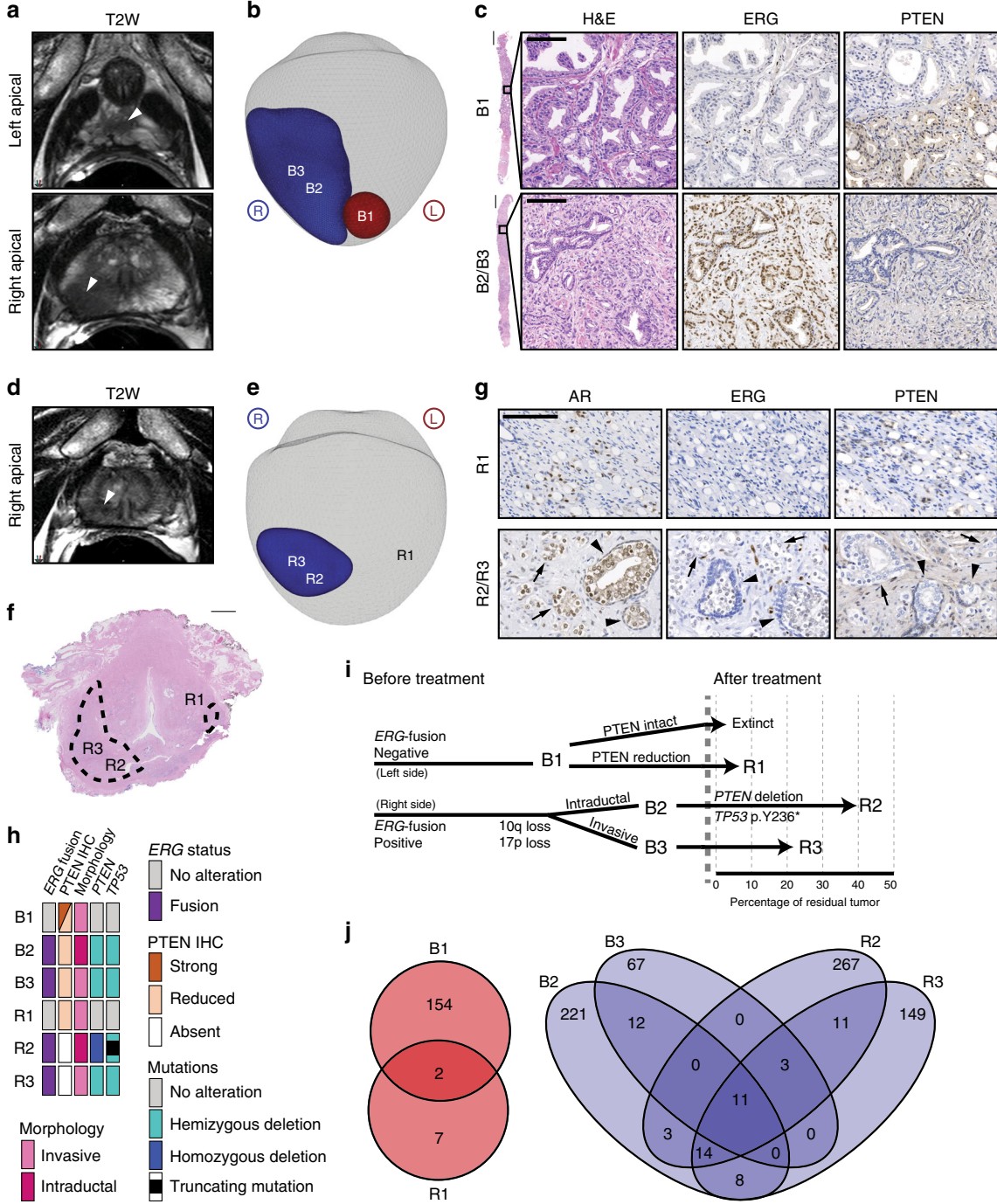

**Fig. 1 Molecular evolution of independent prostate tumor nodules following neoadjuvant intense androgen deprivation therapy. a–g** Pre-treatment (**a–c**) or post-treatment (**d–g**) T2-weighted MRI (**a–d**), volumetric burden (**b–e**), and staining (**c–g**) of a large, semi-contiguous lesion sampled at multiple levels by mpMRI/TRUS fusion-guided biopsy (**c**) or whole-mount pathology (**f, g**) showing the location of biopsied and laser capture microdissected tumor foci. Arrowheads depict lesions on representative slices on MRI (**a, d**) or IDC-P (**g**). Arrows depict invasive carcinoma lesions. Scale bars: whole mount, 5 mm; biopsy, 1000 µm; insets, 100 µm. **h** Oncoprint depicting the alterations and histologic phenotypes of the microdissected tumor foci. **i** Phylogenetic tree depicting the relationships between baseline biopsy and prostatectomy tumor foci with respect to the alterations shown in the Oncoprint. Branches were constructed based on shared and unique mutations in each sample. **j** Venn diagram depicting the overlap between tumor foci based on somatic point mutations and large copy number alterations. Source data for the Venn diagram are provided in a source data file. B2/R2 = baseline/residual intraductal carcinoma, and B3/R3 = baseline/residual invasive carcinoma; B2 was intermingled with B3, and R2 was intermingled with R3.

protocol, and then underwent a second mpMRI prior to RP. The post-treatment mpMRI displayed reductions both to total prostate volume (33 cc to 20 cc) and tumor size (2.8–1.7 cm) (Fig. 1d, e). The right-sided tumor exhibited a partial response to therapy and was visible on both MR imaging and microscopic examination, whereas the left-sided tumor displayed an excellent response and was only detected microscopically (Fig. 1f; Supplementary Fig. 3). Per protocol, the prostatectomy specimen was grossed and sectioned in the same three-dimensional plane as the mpMRI (Supplementary Fig. 3a), and comparison of residual

tumor on the left side revealed that left-sided tumor was likely too hypocellular and scarce to be detected by mpMRI. While diagnostic performance of mpMRI after neoadjuvant treatment is not well-characterized in the literature, mpMRI of untreated prostate cancer is known to have a poor detection rate for clinically insignificant disease[20,21].

Because reconstitution of AR nuclear localization and AR activity are common mechanisms of resistance to neoadjuvant intense ADT[10], we examined nuclear AR expression using immunohistochemistry in treatment-resistant tumor foci in whole-mount RP tissues, as well as anti-PTEN and anti-ERG immunohistochemistry (Fig. 1g; Supplementary Fig. 3b). As the right-sided tumor foci (R2 and R3) displayed reduced but positive ERG staining consistent with suppression of AR activity by ADT plus enzalutamide, we confirmed using immunofluorescence that the right-sided tumor was ERG-positive and distinct from the left-sided tumor (R1), which was ERG-negative (Supplementary Fig. 3c). The left-sided tumor, which showed focal PTEN reduction before treatment, displayed homogeneous PTEN reduction in the minimal residual disease, whereas residual tumor on the right side displayed complete loss of PTEN expression in both IDC-P (R2) and invasive tumor (R3) foci.

Taking the same approach as with the biopsy tissue, we performed LCM and WES on residual tumor foci from the right and left sides, sampling tissue from three separate levels (Supplementary Fig. 3a). Residual IDC-P (depicted as R2) on the right side had two copy losses of PTEN: the same single-copy loss observed on biopsy plus an additional focal loss at 10q23.31 (Supplementary Fig. 2a, b). The residual invasive tumor (R3) on the right side showed the same single-copy loss of PTEN, but no additional loss. Copy number analysis of the residual IDC-P also revealed the same focal loss of chromosome 17p13.1 (Supplementary Table 2) observed on biopsy plus an additional nonsense mutation to TP53 rendering the IDC-P negative for both PTEN and p53. R2 and R3 also harbored the same loss to chromosome 16 detected in the B2 and B3 foci. In contrast, none of these alterations were observed in the residual tumor cells on the left side (R1), which despite being PTEN-deficient by IHC did not harbor any detectable SCNAs (Supplementary Fig. 2a).

**Evolution of treatment-resistant tumor**. In order to visualize the spatial, genomic, and temporal relationships between microdissected pre-treatment biopsies and post-treatment residual tumor, we constructed an oncoprint plot and phylogenetic tree (Fig. 1h, i). Using the TMPRSS2:ERG fusion as a putative early driver of tumorigenesis, based on its expression in all right-sided tumor foci (B2, B3, R2, and R3), we modeled that the IDC-P and invasive foci identified in the RP descended from IDC-P and invasive foci sampled on biopsy. Presumably, a small subpopulation of the IDC-P at baseline (B2) also harbored the second loss to PTEN and the nonsense mutation to TP53, as these were observed in the IDC-P on the right side (R2), which in turn comprised the largest volume of all residual tumor.

Deep focused resequencing of a subset of mutations at ~50,000 × coverage confirmed that of 27 point mutations harbored by R2 and/or R3, no somatic mutations were shared by both R1 and R2 and/or R3 (Supplementary Table 4). Moreover, one point mutation detected in R1 was shared by B1, but neither B2 nor B3, and of six SCNAs observed in R1, one was shared by B1 (Fig. 1j). In addition, 21 mutations were confirmed as shared by B2/R2 and B3/R3. Further strengthening the relationship of the IDC-Pand invasive post-treatment foci to their pre-treatment counterparts was the observation that when point mutations were observed in either B2 or B3, but both R2 and R3 (Supplementary Table 4), the frequency of the mutation

in the matching histologic tumor was generally greater. Combined with shared and distinct somatic copy number alterations (Supplementary Fig. 2a), these data collectively confirm that the right-sided and left-sided residual tumors were clonally independent and descended from the tumor foci sampled on biopsy (Fig. 1j). The absence of any genomic hits to PTEN on the left side suggests alternative mechanisms of suppressing PTEN expression, which may include epigenetic silencing. As all three foci of residual tumor displayed reduced or absent PTEN expression by IHC (Supplementary Fig. 3b), the left- and right-sided tumors independently underwent phenotypic convergence, selecting for subclones with dysregulated PTEN.

## Discussion

Here, we describe a patient with high-risk prostate cancer treated by neoadjuvant intense ADT and surgery. Retrospective analysis of imaging and tissue data revealed that there were two potentially lethal-independent cancers present at baseline, and the two tumors responded differently and demonstrated different status for the tumor suppressor genes PTEN and TP53.

These molecular findings are consistent with large cohort studies of metastatic castration-resistant prostate cancer (mCRPC) previously treated with enzalutamide, in which loss of function of p53 is the most common and enriched alteration relative to untreated primary disease[22,23]. In contrast, PTEN loss was abundant in primary disease and mCRPC, and thus likely contributed to early-stage disease initiation or progression. Treatment of this patient with intense ADT selected for PTEN- and p53-null subclones as they represented the majority of residual disease, with cells harboring epigenetically silenced PTEN or single-copy losses of PTEN and TP53 displaying less of a growth advantage. Meanwhile, the cells with intact PTEN were fully sensitive to intense ADT, as they were not present in the post-treatment specimen.

Multiple lines of histologic and genomic data contributed to the evidence that the left- and right-sided tumors were independent. First, the incidence of the TMPRSS2:ERG fusion is documented to occur early during tumorigenesis, even in precancerous high-grade prostatic intraepithelial neoplasia (PIN), such that it is rarely a subclonal event[24]. We have previously showed that when low- and high-grade tumor foci were sampled from the same index tumor, they were 100% concordant for ERG expression and when positive harbored the same genomic breakpoints[17]. Based on these existing evidence, and morphologic and spatial findings in our study, the left- and right-sided tumors are most likely two juxtaposed independent tumor nodules even though we cannot completely eliminate the possibility that the ERG-positive tumor diverged early from the ERG-negative clone.

Second, our exome sequencing and high-coverage resequencing did not identify somatic variants in common between the left- and right-sided tumor foci, but did confirm the lineage of pre-treatment to post-treatment tumor foci in these tumors. By segmented copy number analysis, the right-sided tumor carried clonal losses to 17p (TP53) and 10q (PTEN), as well as 16q, which carries multiple putative tumor suppressors[25]. In contrast, these were absent from all tumor foci on the left side. The left-sided tumor at baseline harbored a subclonal gain to most of chromosome 7, and these chromosome 7 alterations were absent from all right-sided foci. This chromosome 7 gain may have been associated with the PTEN-intact portion of that left-sided tumor, which completely responded to treatment, potentially explaining in part why it was not detected in the residual tumor from the left side. A limitation of this analysis was the low cellularity of the left-sided residual tumor, which may explain in part why somatic mutations and copy number alterations were infrequent but still

present, distinguishing the sampled cells from adjacent benign tissue.

Third, the ERG-negative and ERG-positive tumors displayed different behaviors in the context of neoadjuvant intense ADT. We observed that the ERG-negative/PTEN-intact subclone was the most sensitive to treatment, and the ERG-positive/PTEN-negative IDC-P subclone was the most resistant in this patient. This pattern is consistent with results from a recent trial of neoadjuvant ADT plus abiraterone and enzalutamide, in which IDC-P, ERG positivity, and *PTEN* loss were associated with the highest residual tumor volumes[7].

Notably, the tumor focus at baseline with IDC-P histology demonstrated the most resistance to therapy, harboring homozygous inactivation of both *TP53* and *PTEN*. Recent reports have also linked prostate ductal and intraductal pathologies to somatic and/or germline alterations affecting the homologous recombination genes *ATM*, *BRCA1*, and *BRCA2*[26–28]. The patient did not carry somatic or germline alterations to these three genes, although the IDC-P focus at baseline did harbor a deletion to chromosome 11 which encompassed *ATM* (see Supplementary Table 2). As this deletion was not observed in its descendent post-treatment tumor (R2), we conclude that the effects of this alteration on *ATM* expression were not required for the maintenance, resistance or progression of the IDC-P tumor focus.

The previous study of neoadjuvant ADT plus abiraterone and enzalutamide also concluded that no tumor with both ERG positivity and *PTEN* loss responded to the point of MRD[7]. MRD has been defined by either RCB <0.25 cc, or as the greatest dimension of residual tumor <0.5 cm. Using this latter criterion, a meta-analysis of three recent trials demonstrated that MRD was a surrogate for 5-year biochemical recurrence-free survival; 11 out of 11 patients in those trials with MRD ≤ 0.5 cm were recurrence-free with median follow-up of 3.4 years[8]. These patients tended to have lower tumor volumes at baseline, but biopsy GG alone did not predict response. In this study, the response of the left-sided tumor met the criteria for MRD despite both the left- and right-sided tumors harboring predominantly Gleason pattern 4 disease.

Integrating the imaging, histologic, and genomic data available to us, we conclude that this patient harbored a polytumor, defined by two completely independent primary tumor systems in the same organ[2]. Evolutionarily, polyclonal or multiple independent prostate tumors trend toward higher risk, because increased diversity of tumor cells increases fitness and thus may increase aggressiveness and treatment resistance[2]. Evolutionary trajectories that resulted in tumor heterogeneity in this case likely contributed to our observation that two independent prostate tumor nodules with distinct genetic alterations responded differently to neoadjuvant intense ADT.

In previous neoadjuvant clinical trials, it was suggested that multiple clones developed in a subset of higher-risk patients prior to therapy, but the absence of targeted biopsies precluded reconstruction of the evolutionary events leading to heterogeneously resistant tumors[9,10]. Our analysis of this patient reveals that genomic profiling of a single biopsy would not have predicted the overall response observed following treatment. It is likely that the spatiogenetic variability of localized prostate tumors will continue to be a major confounder for genetically driven trials in localized prostate cancer, and efforts to integrate multiparametric MRI with histopathology to direct genomic studies in the context of neoadjuvant therapies are needed to characterize independent clones at baseline for optimal clinical management.

## Methods

### Study approval
This clinical and laboratory studies described in this paper were conducted according to the principles of the Declaration of Helsinki. Informed

written consent was received from the study participant prior to inclusion in the study. This clinical study was approved by the National Institutes of Health Institutional Review Board (protocol number 15-c-0124), and was registered with ClinicalTrials.gov (registration number NCT02430480).

### Study subject
The patient is a 66-year-old man who presented to the National Institutes of Health Prostate Multidisciplinary Clinic to discuss treatment options for newly diagnosed high-risk prostate cancer (clinical stage T3a). His PSA had risen slowly over the past 8 years (1.65–5.53 ng/ml), and a templated 6-core biopsy with his primary urologist demonstrated Gleason score 9 and Gleason score 7 cancer. Apical and basal biopsy cores showed evidence of perineural invasion, and conventional imaging (CT, bone scan) did not show any evidence of metastatic disease.

mpMRI of the prostate showed a single semi-contiguous lesion encompassing the right apical–mid peripheral zone also affecting the left distal apical peripheral zone with concern for extraprostatic extension (PIRADS score 5/5), and axial T1-weighted MRI of the abdomen showed no nodal disease. Using right apical and left apical MRI targets, MR/ultrasound-fusion-targeted biopsy of the prostate was performed and confirmed the presence of adenocarcinoma in both lobes: Gleason patterns 4 and 5 in the right apex and right mid regions, and Gleason patterns 3 and 4 in the left apex and right apex regions. Treatment options, including RP and EBRT with ADT, were discussed with the patient along with clinical trial options. The patient enrolled in an IRB-approved phase 2 clinical study of neoadjuvant ADT and enzalutamide (ClinicalTrials.gov #NCT02430480).

The patient was administered two injections of goserelin 12 weeks apart with 160 mg enzalutamide orally daily. After 24 weeks of treatment, his total testosterone declined from 193 ng/dl to 26.8 ng/dl, and his PSA declined from 5.53 ng/ml to <0.02 ng/ml. He then received a second multiparametric MRI, which demonstrated substantial reduction in total prostate volume and a decrease in size of the right apical–mid peripheral zone lesion. No disease was noted on the left side, and no extraprostatic extension or nodal disease was observed on MR. A three-dimensional mold was constructed based on the MR, and following robotic-assisted prostatectomy, the prostate was grossed in the mold to create tissue slices in the same plane as the MRI. Pathologic examination revealed extensive residual disease (cellularity corrected volume of residual cancer burden, RCB) of 0.35 cc on the right side corresponding to the MR-visible lesion, with features of intraductal carcinoma. A much smaller, hypocellular region (RCB 0.01 cc) was also observed on the left side. No additional ADT or RT was performed, and 2 years following surgery, PSA levels remain undetectable.

### Multiparametric MRI acquisition
MR images were acquired at 3-Tesla (Achieva 3.0T-TX, Philips Healthcare, Best, Netherlands) using a combination of the anterior half of a 32-channel cardiac SENSE coil (InVivo, Gainesville, FL, USA) and an endorectal coil (BPX-30, Medrad, Pittsburgh, PA, USA) filled with 45 ml of fluorinert (3 M, Maplewood, MN, USA). The mpMRI protocol included acquisition of T2-weighted (T2W) sequences acquired separately in axial, coronal, sagittal plane, two diffusion-weighted imaging (DWI) sequence acquisitions consisting of one 5 *b*-value acquisition (0–750 s/mm$^2$) for Apparent Diffusion Coefficient (ADC) estimation and one high b-value (2000 s/mm$^2$), and Dynamic Contrast Enhanced (DCE) MRI. DCE MRI images were obtained in 5.6 s intervals following a single dose of gadopentetate dimeglumine 0.1 mmol/kg at 3 ml/s. Axial T1-weighted MR images of the abdomen were also obtained. Full acquisition parameters are listed in Supplementary Table 1.

### Multiparametric MRI interpretation
Pre-treatment mpMRI imaging was interpreted by single expert genitourinary radiologist with more than 10 years' experience (B.T.) following the Prostate Imaging–Reporting and Data System version 2 (PI-RADSv2) guidelines. Five distinct regions within the large apical–mid PZ lesion affecting right side of prostate and extending to left side of the prostate in distal component were identified for biopsy targeting. These five targets included: left distal apical PZ, right distal apical PZ, right posterior apical PZ, right anterior apical PZ, and right mid PZ. Post-treatment imaging was interpreted by the same expert radiologist for assessment of residual tumor burden.

### MR/ultrasound-fusion biopsy
mpMRI was performed upon study entry and again prior to surgery. Biopsy planning and procedure was completed using a commercially available MRI/TRUS fusion-targeting platform (UroNav, Invivo Corp., Gainsville, Florida). MR/ultrasound-fusion biopsies were targeted to regions identified on the first mpMRI. For each MR target, the urologist (P.P.) acquired a biopsy in the axial and sagittal plane of the patient. The axial and sagittal biopsies of the right-sided tumor were concordant for high-grade cancer, perineural invasion, and intraductal carcinoma. Invasive and intraductal tumor foci were separately isolated, and genomic material was pooled across both biopsies by histology. The axial biopsy of the left-sided tumor-only sampled fibromuscular tissue. The sagittal biopsy of the left-sided tumor was used for this study.

### Histology
After sampling, biopsy tissue was placed into formalin and processed into paraffin blocks using standard methods. Following surgery, whole-mount prostate specimens were processed with patient-specific MRI-based 3D-printed

molds for image-pathology correlation[29]. Briefly, the prostate specimen was serially sectioned within the mold from apex to base at 6-mm intervals corresponding to axial planes on mpMRI images. The mold was placed in formalin overnight, the prostate was sliced for whole mounting, and slices were placed in cassettes for processing into paraffin blocks using standard methods.

Biopsy and prostatectomy tissues were sectioned at 5 μm thickness onto charged slides and stained with hematoxylin and eosin using standard methods. Serial sections of biopsy and RP tissues were stained with anti-AR (Cell Signaling, Cat# 5153S, 1:100 into SignalStain diluent), anti-ERG (Abcam, Cat# ab92513, 1:500 into SignalStain diluent), anti-PTEN (Cell Signaling, Cat#9188L, 1:100 into SignalStain diluent), and PIN-4 cocktail (Biocare Medical, Cat# PPM225DS, ready-to-use) antibodies. SignalStain diluent was from Cell Signaling. Slides were baked for 15 min at 60 °C, except for PIN-4 staining which was baked overnight at 45 °C, deparaffinized through xylenes and rehydrated through graded alcohols. Antigen retrieval was performed using a NxGen Decloaker (Biocare Medical), for 15 min at 110 °C in Tris-EDTA (Abcam, Cat# ab93684) for PTEN, and for 15 min at 110 °C in Diva Decloaker (Biocare Medical, Cat# DV2004MX) buffers. Sections were blocked with hydrogen peroxide (Sigma Aldrich, Cat# 216763) for 5 min, blocked with Background Sniper (Biocare Medical, Cat# BS966) for 10 min for PIN-4 or VectaStain Elite ABC HRP kit (Vector Laboratories, Cat# PK-6101) for PTEN, AR, and ERG, and incubated with primary antibody for overnight at 4 °C (AR, ERG, PTEN) or 1 h (PIN-4). Secondary labeling was performed using Mach 2 Double Stain for PIN-4 (Biocare Medical, Cat# MRCT525) or the VectaStain Elite ABC HRP kit for AR, ERG and PTEN for 30 min. Avidin–biotin complexing was then performed for 30 min for AR, ERG, and PTEN. Colorimetric detection was achieved using DAB Peroxidase HRP (Vector Laboratories, Cat# SK4100) for AR, ERG, and PTEN, or Vulcan Red Fast Chromogen (Biocare Medical, Cat# FR805) and Betazoid DAB (Biocare Medical, Cat# BDB2004) for PIN-4. Counterstaining was performed using Mayer's Hematoxylin Solution (Sigma Aldrich, Cat# MHS16). PIN-4 stained slides were air-dried. AR, ERG, and PTEN-stained slides were dehydrated through graded alcohol and cleared in xylenes. Slides were mounted using Permount (Thermo Fisher).

ERG expression of post-treatment specimens was assessed using Opal Multiplex Immunohistochemistry (Akoya Biosciences, Cat# NEL811001KT). Antigen retrieval was performed using a NxGen Decloaker, for 15 min at 110 °C in Opal AR 9 antigen retrieval buffer (Akoya Biosciences, Cat# AR9001KT). Sections were blocked with 2.5% NGS (ImmPRESS HRP Anti-Rabbit IgG polymer kit, Vector Laboratories, CAT# MP-7451) for 10 min, and incubated with anti-ERG (Abcam, Cat# ab92513) primary antibody at 1:500 for 30 min at room temperature. ImmPRESS HRP anti-rabbit IgG polymer was added for 15 min, then an anti-HRP-conjugated Opal 520 fluorophore (Akoya Biosciences, Cat# FP1487001KT) was applied at 1:500 for 10 min. DAPI (4′, 6-diamidino-2-phenylindole, Thermo Fisher, Cat# 1306) was applied at 350 nM for 10 min before slides were mounted using ProLong Glass Antifade Mountant (Thermo Fisher, Cat# P36980).

H&E-, bright-field IHC-, and immunofluorescent IHC-stained slides were scanned using the 20× objective (Plan-Apochromat, NA 0.8) with bright-field or LED fluorescent illumination on an Axio Scan.Z1 (Zeiss) with slide loaders to accommodate standard (25 mm × 75 mm) slides and whole-mount double-wide (50 mm × 75 mm) slides. Immunofluorescent images were uniformly contrast enhanced using ImageJ (NIH). The merged image of immunofluorescence and H&E was generated by removing the background of the fluorescent images in Photoshop CC 2019 (Adobe) and overlaying onto the bright-field image.

**Residual cancer burden calculations.** Residual tumor was concordantly identified by three board-certified genitourinary pathologists (M.M., R.L., and H.Y.). Residual cancer burden (RCB)[30] was measured and calculated by multiplying the number of slices through which each residual tumor extended by the largest cross-sectional width and length and block thickness (0.6 cm). Volume was further corrected by multiplying by 0.4 and the estimated tumor cellularity. Precise length and width measurements were performed on scanned slides using Zen Blue 2012 (Zeiss) with objective/magnification and pixel:distance calibrations recorded within the scanned CZI file. The right-sided lesion extended through four blocks. With a length of 2.09 cm, a width of 1.73 cm, and 10% cellularity, the RCB was 0.35 cc. The left-sided lesion extended through two blocks. With a length of 0.62 cm, a width of 0.35 cm, and 10% cellularity, the RCB was 0.01 cc.

**Genomic analysis.** Regions of residual tumor identified in whole-mount prostatectomy specimens were registered with post-treatment mpMRI, and co-registered with pre-treatment mpMRI using imaging landmarks for reference. Biopsy paths were superimposed on pre-treatment MR targets to indicate the spatial relationship between biopsy tissue and RP tissue. Serial sections of tumor tissue (and benign regions uninvolved with tumor) were cut onto metal frame PEN-membrane slides (MicroDissect GmbH), stained with Paradise Plus stain (Thermo Fisher), and laser capture microdissected using an ArcturusXT Ti microscope (Thermo Fisher) onto CapSure Macro LCM Caps (Thermo Fisher). Serially stained slides of PIN-4, anti-AR, and anti-ERG immunostaining were used as references.

Using the QIAamp DNA FFPE Tissue Kit (Qiagen), microdissected cells adhered to caps were lysed and DNA was purified according to the manufacturer's instructions. DNA yields were quantified using Picogreen reagent (Thermo Fisher). For exome sequencing, 10–50 ng of DNA were sheared (Covaris), and assembled into whole-exome libraries using the SureSelect Human All Exon V7 Low Input Exome (Agilent). Samples were sequenced to an average depth of 35× on-target coverage on an Illumina HiSeq4000 at 150 cycles paired-end, eight indexing cycles, and ten molecular barcoding cycles. Pass-filter FASTQ files were trimmed using SureCall Trimmer 4.0.1 (Agilent), aligned with the Burrows Wheeler Aligner BWA-MEM version 0.7.17[31] to version hg19 of the human genome (b37 with decoy chromosomes), duplicate-marked using LocatIt 4.0.1 (Agilent), and quality score recalibrated using version 4.0.5.2 of the Genome Analysis Toolkit (GATK). Somatic point mutations were called on intervals from the Agilent bait design BED files using MuTect2 (part of the GATK4 package), first by running MuTect2 in tumor-only mode on all BAM files generated by our group using the same Agilent library preparation method from normal (not tumor) FFPE DNA and merging the resultant VCF files using the CreateSomaticPanelOfNormals module of GATK. MuTect2 was run in somatic mode on each tumor BAM paired with the normal BAM from this patient and the somatic panel of normal with af-of-alleles-not-in-resource set to 0.0000025 disable-read-filter set to MateOnSameContigOrNoMappedMateReadFilter. GetPileupSummaries and CalculateContamination were used on each tumor BAM file, and the resultant contamination table was used to filter somatic mutations using FilterMutectCalls. CollectSequencingArtifactMetrics and FilterByOrientationBias were used to further filter mutations for 8-oxoG artifacts using the settings -AM G/T -AM C/T. These pass-filter mutations were then functionally annotated using Oncotator version 1.9.70 (database version April052016). Mutations were then further filtered on a per-sample basis by requiring any locus with a somatic mutation be covered at least half of the mean coverage for the sample and by no fewer than 16 reads and at least 2 unique counts of the alternate allele. Mutations resulting from obvious read bias were removed by running HaplotypeCaller on the reassembled BAM output of MuTect2 in somatic mode and using the directionality of the reads in the strand direction-annotated VCF file. All remaining pass-filter SNVs and indels were then evaluated against the output of MuTect2 from the other samples. For loci where mutations were called confidently in one sample but filtered out, those mutations were backfilled. For loci where no mutations were called, the read depth of the reference allele was backfilled using Pysamstats version 0.24.3 in Python 2.7.

Somatic copy number alterations and aCGH-style plots were generated using Nexus Copy Number version 9 (BioDiscovery). All BAM files generated by our group using the same Agilent library preparation method from normal (not tumor) FFPE DNA were loaded into the Multiscale Reference Builder, specifying the Agilent bait design BED file for selecting regions of the genome to estimate copy number. All default settings were used. After generating the pooled reference, each tumor BAM file was separately loaded with the following analysis settings: systematic correction: quadratic correction; recenter probes: median; analysis: SNP-FASST2 segmentation; significance threshold: 1.0E-6; max contiguous probe spacing (Kbp): 1000.0; percent outliers to remove: 3.0. Because Nexus copy number relies on arbitrary cutoffs for calling gains and losses, we imputed SCNA calls from GATK4 for determining whether deviation from a copy number ratio of 0 was a meaningful alteration. To restrict analysis to only those regions covered by the exome library, the Agilent library design BED file was preprocessed with PreprocessIntervals with bin-length set to 0 and interval-merging-rule set to OVERLAPPING_ONLY. The BED file was also annotated with GC content using AnnotateIntervals. These interval files were used in all GATK4 copy number calling steps. To obtain SCNA calls with GATK4, read counts were first obtained from all BAM files using CollectReadCounts, including from all BAM files generated by our group using the same Agilent library preparation method from normal (not tumor) FFPE DNA. The normal read count files were compiled into a panel of normal using CreateReadCountPanelOfNormals, and the panel of normal was used to smooth read counts across all samples using DenoiseReadCounts. CollectAllelicCounts was also performed on all denoised BAM for identifying regions of potential LOH. ModelSegments used tumor smoothed read counts and the paired normal/tumor allelic counts for generating copy number estimates that were then called using CallCopyRatioSegments.

To determine whether an unbiased assessment of all SNVs and copy number alterations would identify clonal somatic events based on grouped analysis of all six tumor samples, we employed CLONET (version 1.0.0 revision 20171016) with the v2 method for estimating error[32]. SCNAs called by both Nexus and GATK were reformatted into SEG files, and each tumor BAM file was filtered for informative SNPs using ASEQ version 1.1.11. We then parsed the point mutation and SCNA clonality tables for events marked "clonal" or "subclonal" by CLONET, which would indicate that CLONET had identified molecular evidence of a single tumor cell population with expected allelic fractions in interrelated subclones. However, CLONET marked all events as "subclonal.uncertain" or "not.analysed" which occurs when segments and mutations do not fit the assumption of the expected model for identifying subclonal populations[32].

Based on pass-filter SNVs and indels identified from exome sequencing, 50 targets were selected randomly from 871 total calls for additional amplicon-based sequencing, with the exception of the TP53 Y236X nonsense alteration, which was subsequently included for verification[33,34]. Prior to nomination for targeted sequencing, each selected genomic coordinate was inspected in IGV and mutation calls falling in MAPQ < 10 were excluded. Four of the randomly selected mutations were excluded by this vetting process, resulting in 47 total targets. Using parameters to limit the amplicon size to 140 nucleotides, we used the Ion AmpliSeq Designer [https://www.ampliseq.com] tool which identified 46 multiplex primer pairs that were compatible with our intended targets in a single pool. All multiplex

primer panel designs were cross-checked using In-Silico PCR [https://genome.ucsc.edu/cgi-bin/hgPcr] to prevent production of unintended amplicons for each patient-specific batch of targets. The forward and reverse primer sequences identified above were appended with adaptor sequences[33] at the 5′ end, complementary to Illumina Nextera full-length primers (IDT). Dual-barcoded sequencing adaptors were modified[33] to contain longer regions of complementarity and were ordered with NGSO-4 purity from Sigma. Primers were resuspended in 1 × TE buffer (pH 8.0, 100 μM). A designated pre-PCR workstation was used for the first reaction. A mixture of all forward primers was prepared such that each primer was represented at 2 μM concentration. An aliquot of the primer mix (2.5 μL) was combined with uracil N-glycosylase (0.3 U, 2 μL), cfDNA (8 μL), 2 × KAPA HiFi HotStart Uracil + ReadyMix (15 μL), and dUTP (3.5 μL, 2.5 μM). Samples were incubated in a thermal cycler at 37 °C for 2 min, 98 °C for 5 min, 65 °C for 2 min, and 72 °C for 7 min. Products from this reaction were cleaned using AMPure XP SPRI beads (54 μL for each 30 μL tagging reaction), following the manufacturer's recommended protocol for mixing, magnetic separation, ethanol washes, and drying. Elution was performed with deionized nuclease-free water (10 μL) to recover the purified tagged product. Purified tagged products (8 μL) were combined with ddATP (2 μL, 0.2 mM), 10 × CoCl₂ (2 μL), 10 × terminal deoxynucleotidyl transferase (Tdt) buffer (2 μL), low TE buffer (4.4 μL), and Tdt enzyme (1.6 μL of 20 U) for a total volume of 20 μL. Samples were incubated 90 min at 37 °C and 15 min at 75 °C. Samples were held at 4 °C until clean-up, which was performed using the Select-A-Size Clean & Concentrator Kit (Zymo) according to the manufacturer's instructions for a 100 bp one-sided cutoff with the following modifications: the last wash step was centrifuged at maximum speed for 1 min, and elution was performed with 10 μL Zymo elution buffer with a 5 min incubation. 8 μL of purified product were recovered. For the amplification of targets using the reverse primers, a mixture of reverse oligonucleotides was created with 1 μL of each 100 μM primer in a total volume of 100 μL TE, for a final concentration of 1 μM each primer. To 1.25 μL of this primer mix, 3.5 μL of 2.5 μM dUTP, 1 μL of TE, 8 μL of product from the terminal transferase reaction, 1.25 μL of a 2 μM stock of i5 barcoded sequencing adaptor[33], and 15 μL of 2 × KAPA HiFi HotStart Uracil + ReadyMix were added. Samples were incubated in a thermal cycler at 98 °C for 5 min initially and followed by 20 cycles of PCR (98 °C–65 °C–72 °C for 30 s each with 7 min of final extension at 72 °C). Clean-up was performed using the Select-A-Size Clean & Concentrator Kit using the guidelines for a 200 bp one-sided cutoff according to the manufacturer's protocol with the following modifications: the last wash step was centrifuged at maximum speed for 1 min, and elution was performed with 10 μL Zymo elution buffer following a 5 min incubation. In total, 8 μL of purified product were recovered. Final construction of each sequencing library was performed by additional amplification using both the NGSO-4 i5 and i7 sequencing adaptors to prime PCR. All 8 μL of product from the previous step was mixed with 4 μL of 4 μM i7 adaptor stock, 4 μL of 2 μM i5 adaptor stock, 4 μL of TE, and 20 μL of 2 × KAPA HiFi HotStart Uracil + ReadyMix. Samples were incubated in a thermal cycler at 98 °C for 5 min and followed by 10–30 cycles of PCR (98 °C–72 °C for 30 s each with 7 min of final extension at 72 °C). The exact number of cycles of PCR to use were determined empirically by substituting control FFPE genomic DNA in the first step of the library preparation protocol, substituting 4 μL of 6 × SYBR green in the final PCR step above, and performing qPCR on a QuantStudio 3 (Thermo Fisher). The number of cycles was determined by cycle threshold value corresponding to the ½-maximum of amplification saturation on a linear curve. Clean-up was performed using the Select-A-Size Clean & Concentrator Kit with a 200 bp one-sided cutoff as described above, but with a 20 μL elution. Quality control was performed on the completed library using D1000 ScreenTapes on a TapeStation looking for a peak size between 250 bp and 350 bp with primer dimer comprising <10% of the library. The library as then quantified for functional concentration using the KAPA Illumina Quantification Kit for NGS, using a 1:50,000 dilution of the library per the manufacturer's protocol. The library was adjusted to a final concentration of 10 μM.

The amplicon library was sequenced on a MiSeq (Illumina) instrument with the v2 Reagent Kit (500-cycle). 20% Phi-X was added to the final library pool. Sequencing was performed with 195 cycles paired-end and dual 8-cycle indexing reads on the P5 and P7 adaptors. This sequencing strategy intentionally sequenced through the entire insert molecule and into the adaptor on the other side, reading the i5 and i7 indexing adaptor an additional time. Individually barcoded libraries were recovered and filtered for quality using bcl2fastq (Illumina). Pass-filter FASTQ files were trimmed with Trimmomatic version 0.36, preserving the trimmed-off sequences and parsing them for the expected i5 and i7 adaptor based on bcl2fastq binning. Comparing each library to the expected adaptor pair, read pairs corresponding to the incorrect adaptors were discarded. Remaining read pairs in FASTQ files were aligned to b37-decoy using BWA-MEM version 0.7.10. BAM files were sorted and indexed using samtools version 1.9, and unique base calls were identified by running using Pysamstats version 0.24.3 in Python 2.7 over the defined intervals for each target, and all calls were manually inspected using IGV version 2.4.19.

Of the 46 targets, 1 target was in a region of poor complexity and was removed from future consideration. One target failed to amplify in all samples, while five targets preferentially demonstrated bias toward the alternate allele in the normal sample and were also removed. Based on counts of the alternate allele determined a priori from exome sequencing, a sample was determined to harbor the mutation in

question if (1) the frequency of the mutation in the tumor sample was three times greater than in the benign reference; (2) if the alternate absolute allele read count was greater than 50; and (3) if the read count for the base in question was at least three times greater than the average alternate allele for the remainder of the locus (i.e., above the background error rate of library preparation and sequencing).

**Reporting summary**. Further information on research design is available in the Nature Research Reporting Summary linked to this article.

## Data availability

For the protection and privacy of the human subject in this study, the raw whole-exome and targeted sequencing data have been deposited in NCBI Database of Genotypes and Phenotypes (dbGaP), a controlled-access database, under the accession ID phs001938.v1.p1. All the other data supporting the findings of this study are available within the article and its supplementary information files, and from the corresponding author upon reasonable request. A reporting summary for this article is available as a Supplementary Information file. The source data underlying Fig. 1j, Supplementary Table 3, and Supplementary Table 4 are provided as a Source Data file.

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

## Acknowledgements

The authors gratefully acknowledge the patient and the family of the patient who participated in this study. DNA sequencing was performed at the Center for Cancer Research (CCR) Genomics Core, the CCR Illumina Sequencing Facility, and the CCR Genomics Technology Laboratory. The authors acknowledge technical assistance from Steven Shema and Madeline Wong. Portions of this work utilized the computational resources of the NIH HPC Biowulf cluster. This work was supported by the Prostate Cancer Foundation (Young Investigator Awards to S.W., S.H., F.K., D.J.V., R.A.M., H.Y., and A.G.S.), the Department of Defense Prostate Cancer Research Program (W81XWH-19-1-0712 to S.W., W81XWH-16-1-0433 to A.G.S.), the National Cancer Institute (HHSN261200800001E) and the Intramural Research Program of the NIH, National Cancer Institute.

## Author contributions

S.W., S.H., H.Y., B.T., and A.G.S. designed the research study. P.A.P., P.L.C., and W.L.D. designed the clinical study. F.K., P.A.P., R.A.M., D.J.V., G.C., J.L.G., M.J.M., and B.T. were involved in patient care and acquired patient data. S.W., D.J.V., R.T.L., R.L., R.A., J.R.B., N.V.C., S.Y.T., M.J.M., H.Y., and A.G.S. performed histopathological analyses. S.H. and B.T. performed imaging analyses. S.W., N.T.T., S.Y.T., and A.G.S. performed genomics analyses. S.W., S.H., D.J.V., H.Y., R.T.L., B.T., and A.G.S. interpreted results. S.W., S.H., and A.G.S. prepared figures. S.W., S.H., N.V.C., H.Y., and A.G.S. drafted the paper. All authors approved the final paper.

## Competing interests

H.Y. and R.T.L. perform consulting in an advisory role for Janssen Pharmaceuticals. The remaining authors declare no competing interests.
