## [Peer Review File · Nature Communications]

Reviewers' comments:

Reviewer #1 (Remarks to the Author):

Wilkinson and colleagues have performed an exhaustive analysis of a single localized prostate cancer case. The case itself is remarkable: a newly-diagnosed patient treated with a super-castration protocol, leading to partial response. That would be unremarkable by itself, but here the authors show that the origins of this partial response are two completely separate cancers, one highly-responsive to ADT and only generally non-responsive. Poly-tumours have been posited in prostate cancer for decades, and were established firmly a few years back. This is the first demonstration that poly-tumours (as opposed to poly-clonal tumours) can actually have direct therapeutic consequences, and thus is of major potential impact. Technically it is well-executed, and the manuscript is tightly written and an easy read.

1. Sensitivity

The sequencing depth is reported as $\sim 35x$, is this sufficient for verifying poly-tumour behaviour. It seems prudent to sequence this single case significantly deeper ($\sim 120x$) to verify the correct evolutionary application. Targeted sequencing approaches could also be used, but may be less useful for a single-case study.

2. Methods

Please give the software versions and parameters for all analysis tools used, and indicate the repository and accession used for data deposition.

3. Central Review

It might be valuable for such a single-case study to have multiple pathologists and multiple radiologists review the H&Es and mpMRI scans respectively.

Reviewer #2 (Remarks to the Author):

This is an interesting study based on a neoadjuvant ADT/enzalutamide 'trial' of one 66-year old prostate cancer (PCa) patient who was diagnosed of a large semi-contiguous localized adenocarcinoma. The central message of "Multiple primary tumors with differential drug sensitivity" has clinical implications for understanding clonal tumor cell evolution and for developing personalized therapies against different tumor clones with divergent genetic/genomic alterations.

MAJOR:

1. The authors ought to link differential ADT/enzalutamide sensitivities to genetic/genomic alterations rather than descriptive morphological classifications (e.g., invasive) of primary tumors, and this concept needs to be clearly relayed in the Title, Abstract, Text, and Conclusions. The pre-treatment lesions, B1 (L) and B2/B3 (R) are clearly distinct with respect to their genomic changes, which directly correlate with their therapy response. Specifically, the B1 tumor overall presented as more differentiated adenocarcinomas, lacked ERG overexpression (presumably lacked the TMPRSS/ERG fusion?), and was heterogeneous in PTEN IHC positivity. Strikingly, the part of B1 tumor with intact PTEN was eradicated by goserelin/enzalutamide whereas the part of B1 tumor with PTEN reduction evolved into a resistant (R1) tumor. In contrast to B1 tumor, the B2/B3 tumors clearly presented as largely undifferentiated and more aggressive tumors, had positive ERG IHC, showed 10q (PTEN) loss and/or 17p (p53) loss/mutations, and were 'intrinsically' resistant to goserelin/enzalutamide. These results are overall consistent with recent studies demonstrating that genetic loss of PTEN and p53 and/or RB1 causes PCa cell lineage plasticity and confers resistance to AR-targeting therapies.

2. In fact, the morphological descriptions of the B1 vs. B2/B3 tumors were confusing, inconsistent, inaccurate, and likely incorrect. As mentioned above, the IHC images (Fig. 1c) to this reviewer

(who was trained as a pathologist) clearly showed that the B1 tumor was more differentiated whereas the B2/B3 tumors less differentiated and more aggressive. However, the authors described the B1 tumor as '..... the invasive lesion.....' (Abstract). Also, in 'Methods' under "Study Subject and Clinical History', the left-side (B1) tumors were described as Gleason patterns 4 and 5 whereas the right-side (B2/B3) tumors 'Gleason patterns 3 and 4', which do not match (are actually opposite to) the histological presentations shown in Fig. 1c.

3. Results in Fig. 1g showed negative ERG IHC in post-treatment refractory (R) tumors but the Text said '..... reduced but positive ERG staining,.....' (page 5; top paragraph). Why? How did the resistant tumors become ERG-negative? Does this mean that ERG was not upregulated in primary B2/B3 tumors via genomic fusions but through some epigenetic mechanisms?

MINOR:

The manuscript format is odd and inconsistent with the journal. There is also some strange wording (e.g., '..... terminally differentiating transcription factor.....'; page 3) and grammatical errors.

Reviewer #3 (Remarks to the Author):

In the submitted manuscript the authors present an n=1 study of a 66 yo patient with a high risk localized prostate cancer enrolled in a clinical trial with ADT + Enzalutamide in the neoadjuvant setting.

The authors identified two spatially separated primary tumours and went on to perform limited molecularly characterization of the lesions before and after treatment. Overall, the manuscript is well written.

Major comments:

1) The main concern is the limited data provided and consequently, lack of novelty. This is limited to H&E, ERG IHC, PTEN IHC. Papers using these techniques have been published for >10 years - the novelty is sampling before and after Neo-adjuvant treatment but given the authors have assessed all lesions before and after treatment using whole exome sequencing, I was interested in a much more comprehensive and thorough analysis. at a minimum one would like to see mutation calls and the copy number profiles. this is especially necessary given the authors' claim that the lesions are clonally independent. Without sufficient data, their statement is not enough supported.

Multifocality of prostate cancer has been widely described and recently McKay RR et.al (J Clin Oncol. 2019, ref 9) showed PTEN and ERG status post neoadjuvant enzalutamide. So, each part has been described previously.

Minor comments:

1) A clear disadvantage of WES (vs WGS) is that the authors may miss rearrangements and other structural variations that could potentially allow a more accurate interpretation of the clonal similarities or dissimilarities between the lesions.

2) As intraductal carcinomas are known to be related to DNA repair genes defects it would be interesting to report/discuss its BRCA/ATM status.

3) In the figure we would suggest moving some of the IHC images to supplementary and give more space for molecular data of interest. i would also suggest putting R1 next to R1 and so forth to link it better with figure i.

Response to reviewers' comments.

The authors are grateful for the reviewers for their critical and thoughtful critique of our manuscript. We have made point-by-point responses (shown in bold) as we have attempted to address each concern.

Reviewer #1 (Remarks to the Author):

Wilkinson and colleagues have performed an exhaustive analysis of a single localized prostate cancer case. The case itself is remarkable: a newly-diagnosed patient treated with a super-castration protocol, leading to partial response. That would be unremarkable by itself, but here the authors show that the origins of this partial response are two completely separate cancers, one highly-responsive to ADT and only generally non-responsive. Poly-tumours have been posited in prostate cancer for decades, and were established firmly a few years back. This is the first demonstration that poly-tumours (as opposed to poly-clonal tumours) can actually have direct therapeutic consequences, and thus is of major potential impact. Technically it is well-executed, and the manuscript is tightly written and an easy read.

We agree that our observation that two independent primary cancers of the same organ (a polytumor) responded differently to the same treatment may be clinically meaningful and we have strived to make this manuscript short and accessible. As we note below in comments to other authors, we have revised and expanded the manuscript in response to editorial considerations but also to include additional data requested by all reviewers. We have provided these data as Supplementary figures and tables, so as not to disrupt the ease of navigating a 1-figure paper. However, if the reviewers strongly feel that these additional data justify including them as main text display items, we will make the appropriate revisions.

1. Sensitivity

The sequencing depth is reported as ~35x, is this sufficient for verifying poly-tumour behaviour. It seems prudent to sequence this single case significantly deeper (~120x) to verify the correct evolutionary application. Targeted sequencing approaches could also be used, but may be less useful for a single-case study.

While our original sequencing depth of ~35x allowed us to reliably resolve recurrent somatic copy number events (such as those shared between related tumor foci), it was not sufficient for conclusive determination of clonal architecture as can be measured by point mutations. Indeed, the majority of our original conclusions were based on shared or mutually exclusive copy number calls. The high tumor purities afforded by the laser capture microdissection we performed certainly added confidence to our existing calls. Nonetheless, and the reason why our revision has been delayed, was that we attempted to perform additional whole exome sequencing to resolve this concern. First, we attempted to resequence our existing libraries, and although we achieved a raw depth of >1000x, after marking duplicates we did not gain any additional unique depth. This speaks to the technical challenges associated with performing whole exome sequencing in this setting, with low input amounts from laser capture microdissection leading to limited library complexity. We did not have any additional DNA from these samples, so we performed additional laser dissections of new tumor foci from the limited amount of material available to us, and we attempted new library generation. This too was not successful at achieving greater sequencing depth than our original attempt, especially since we used most of the available material the first time. However, we saved a portion of the DNA from this second extraction to perform targeted sequencing as suggested by Reviewer 1. These data, which are sequenced to an average depth of ~50,000x coverage per amplicon, reliably demonstrated a relationship between all foci on the right side and that somatic mutations from the right-sided tumor were mutually exclusive to tumor foci on the left side. More specifically, this additional resequencing confirmed the exclusivity of the TP53 mutation to the R2 focus. In cases where our sampling of residual tumor in R2 or R3 resulted in cross-contamination due to the technical challenges of laser microdissection (i.e. IDC-P tumor cells in the invasive pool or vice versa), we note that where a mutation was strongly enriched in the B2 focus, its overall frequency was greater in R2 than in R3. We showed the same general effect with B3 mutations and their detection in R3 versus R2. (Our raw counts are provided as source data.)

2. Methods

Please give the software versions and parameters for all analysis tools used, and indicate the repository and accession used for data deposition.

We have substantially revised the methods section to include software versions and parameters used for our analyses. We have registered our study and are depositing our data into dbGaP, accession ID phs001938.v1.p1. The data availability statement in the main text has been updated with this information.

3. Central Review

It might be valuable for such a single-case study to have multiple pathologists and multiple radiologists review the H&Es and mpMRI scans respectively.

When the clinical trial for which this patient enrolled was initiated, there was no central review of pathology or imaging as part of the trial. Nonetheless, in addition to the pathologist who read all slides on the study (Dr. Merino) we have collaborated with two additional GU pathologists with expertise in reading slides following intense neoadjuvant ADT (Dr. Lis and Dr. Ye). All three pathologists have concurred on the independence of the left- and right-sided tumors based on immunohistochemistry and tissue morphology. We further consulted with the NCI radiologist, Dr. Turkbey, and collaborated with an additional data scientist with expertise in interpreting MRI, Dr. Harmon. Collectively, it was determined that additional review of mpMRI by more than one radiologist would not be informative, especially since there are no other radiologists (to the best of our knowledge) at centers that perform a second mpMRI after neoadjuvant ADT prior to surgery.

Reviewer #2 (Remarks to the Author):

This is an interesting study based on a neoadjuvant ADT/enzalutamide 'trial' of one 66-year old prostate cancer (PCa) patient who was diagnosed of a large semi-contiguous localized adenocarcinoma. The central message of "Multiple primary tumors with differential drug sensitivity" has clinical implications for understanding clonal tumor cell evolution and for developing personalized therapies against different tumor clones with divergent genetic/genomic alterations.

So that there is no misunderstanding, this case is from a larger trial of 39 patients for which a manuscript reporting the primary outcomes is in preparation. We have updated our manuscript to make this clearer. We believe that our unique observation of two independent tumors from a single patient would be a "footnote" in the larger manuscript, which is why we prepared the current n of 1 report. At the time of our initial submission, that trial had concluded enrollment but was still in progress. At the time of this revision, that trial is complete.

MAJOR:

1. The authors ought to link differential ADT/enzalutamide sensitivities to genetic/genomic alterations rather than descriptive morphological classifications (e.g., invasive) of primary tumors, and this concept needs to be clearly relayed in the Title, Abstract, Text, and Conclusions.

As the genomic alterations are a major finding from this effort, we agree that making the genomic alterations clearly linked to resistance is important. However, it is not possible to include this level of detail in a title limited to 12 words. We have revised the abstract, results and conclusions per this recommendation.

The pre-treatment lesions, B1 (L) and B2/B3 (R) are clearly distinct with respect to their genomic changes, which directly correlate with their therapy response. Specifically, the B1 tumor overall presented as more differentiated adenocarcinomas, lacked ERG overexpression (presumably lacked the TMPRSS/ERG fusion?), and was heterogeneous in PTEN IHC positivity. Strikingly, the part of B1 tumor with intact PTEN was eradicated by goserelin/enzalutamide whereas the part of B1 tumor with PTEN reduction evolved into a resistant (R1) tumor. In contrast to B1 tumor, the B2/B3 tumors clearly presented as largely undifferentiated and more aggressive tumors, had positive ERG IHC, showed 10q (PTEN) loss and/or 17p (p53) loss/mutations, and were 'intrinsically' resistant to goserelin/enzalutamide. These results are overall consistent with recent studies demonstrating that genetic loss of PTEN and p53 and/or RB1 causes PCa cell lineage plasticity and confers resistance to AR-targeting therapies.

Clearly, lineage plasticity in prostate cancer represents a spectrum of heterogeneity, where aggressive variant prostate cancer (Aparicio et al Clinical Cancer Research 2016) and AR-high prostate cancer (Labrecque et al JCI 2019) are on nearly opposite ends of the spectrum. In this case, plasticity is clearly manifested by dedifferentiation but limited to the extent where we show the residual tumor is still positive for nuclear AR.

2. In fact, the morphological descriptions of the B1 vs. B2/B3 tumors were confusing, inconsistent, inaccurate, and likely incorrect. As mentioned above, the IHC images (Fig. 1c) to this reviewer (who was trained as a pathologist) clearly showed that the B1 tumor was more differentiated whereas the B2/B3 tumors less differentiated and more aggressive. However, the authors described the B1 tumor as '..... the invasive lesion.....' (Abstract). Also, in 'Methods' under "Study Subject and Clinical History", the left-side (B1) tumors were described as Gleason patterns 4 and 5 whereas the right-side (B2/B3) tumors 'Gleason patterns 3 and 4', which do not match (are actually opposite to) the histological presentations shown in Fig. 1c.

Thank you for pointing out this error. This was a mistake on our part in compiling the manuscript in which we switched the left and right descriptions when assembling the clinical history portion of the methods section. This has been corrected in our revision.

3. Results in Fig. 1g showed negative ERG IHC in post-treatment refractory (R) tumors but the Text said '..... reduced but positive ERG staining,....' (page 5; top paragraph). Why? How did the resistant tumors become ERG-negative? Does this mean that ERG was not upregulated in primary B2/B3 tumors via genomic fusions but through some epigenetic mechanisms?

We appreciate the confusion. As an AR-regulated gene, TMPRSS2 expression decreases in the context of ADT. Consequently, immunodetection of the TMPRSS2-ERG fusion would similarly be reduced in fusion-positive cases treated with intense ADT. We attempted to illustrate the difference between ERG-positive and ERG-negative treatment refractory tumors in our previous figure, and while the difference is apparent on our monitors when looking at large areas of tissue, it was not obvious in our snapshots. Therefore, we have taken two methods to resolve this. First, we performed uniform contrast enhancement on regions of post-treatment tumor where we stained with anti-ERG. This change is reflected in new micrographs for Figure 1g and in Supplementary Figure 3. In addition, we used an FFPE-compatible immunofluorescent technique, Opal labeling (from Akoya), with our anti-ERG antibody. Again, using uniform background adjustment the difference is quite obvious between R1 and R2/R3 regions (Supplementary Figure 3). After scanning, we restained the slide with H&E to overlay the ERG-positive or negative tumor cells over brightfield visible regions of tumor. Please also note that ERG expressed in endothelial cells is not influenced by ADT, and this normal staining can be seen in both the R1 and R2/R3 regions.

MINOR:

The manuscript format is odd and inconsistent with the journal. There is also some strange wording (e.g., '..... terminally differentiating transcription factor.....'; page 3) and grammatical errors.

We have revised the manuscript in accordance with Nature Communications for matting requirements and performed additional proofreading.

Reviewer #3 (Remarks to the Author):

In the submitted manuscript the authors present an n=1 study of a 66 yo patient with a high risk localized prostate cancer enrolled in a clinical trial with ADT + Enzalutamide in the neoadjuvant setting.

The authors identified two spatially separated primary tumours and went on to perform limited molecularly characterization of the lesions before and after treatment. Overall, the manuscript is well written.

Major comments:

1) The main concern is the limited data provided and consequently, lack of novelty. This is limited to H&E, ERG IHC, PTEN IHC. Papers using these techniques have been published for >10 years - the novelty is sampling before and after Neo-adjuvant treatment but given the authors have assessed all lesions before and after treatment using whole exome sequencing, I was interested in a much more comprehensive and thorough analysis. at a minimum one would like to see mutation calls and the copy number profiles. this is especially necessary given the authors' claim that the lesions are clonally independent. Without sufficient data, their statement is not enough supported.

As described above in our response to Reviewer 1, we have provided summary and raw data for our mutation calls and copy number profiles in supplementary and source data. We believe that with our original and additional analyses the evidence now supports our original conclusion.

Multifocality of prostate cancer has been widely described and recently McKay RR et.al (J Clin Oncol. 2019, ref 9) showed PTEN and ERG status post neoadjuvant enzalutamide. So, each part has been described previously.

While heterogeneity and PTEN/ERG status are documented in prostate cancer, the clonal independence of the tumors at baseline and post-treatment has not been described before and has implications for clinical management.

Minor comments:

1) A clear disadvantage of WES (vs WGS) is that the authors may miss rearrangements and other structural variations that could potentially allow a more accurate interpretation of the clonal similarities or dissimilarities between the lesions.

We agree and regret that WGS on FFPE samples is still not feasible. The required input amounts (given that the DNA is degraded) would easily exceed what can be reasonably microdissected from these tissues. Consequently, we have made sure in our manuscript not to imply that we have fully determined the entire spectrum of genomic alterations that may drive progression, resistance, or sensitivity of the different clones. There is clearly much that we miss with short read sequencing technology.

2) As intraductal carcinomas are known to be related to DNA repair genes defects it would be interesting to report/discuss its BRCA/ATM status.

We have included this analysis in Supplementary data and in our discussion. There were no consistent alterations to BRCA1, BRCA2 or ATM that would indicate they were driving the IDC-P phenotype.

3) In the figure we would suggest moving some of the IHC images to supplementary and give more space for molecular data of interest. I would also suggest putting R1 next to R1 and so forth to link it better with figure i.

For this revision, we have attempted to balance Reviewer 1's comment that "manuscript is tightly written and an easy read" as a short, 1-figure paper with editorial requirements and Reviewer 2's note that the "manuscript format is odd and inconsistent with the journal." Consequently, we have modestly expanded the results and discussion section of the paper, keeping it as 1 figure with new supplementary figures and tables. If the reviewers and/or editors would prefer us to modify the paper further, we will.

REVIEWERS' COMMENTS:

Reviewer #1 (Remarks to the Author):

All concerns fully addressed, and the reviewer appreciates the care taken in addressing the challenges of working with limited clinical trial material.

Reviewer #2 (Remarks to the Author):

This is a revised and significantly improved version of the original manuscript reporting a single case of a prostate cancer patient who went through extensive neoadjuvant ADT (goserelin and enzalutamide) prior to RP. The authors have conscientiously addressed my questions. This reviewer has identified only a few minor mistakes/errors in the writing.

1) Supplementary Figure 1 legend: c, should be "...left-sided B1 and right-sided B2/B3 biopsies....".

2) Line 150 beginning: "and" should be "an".

3) Line 356: "... The prostate gland was specimen was serially...." does not make sense and perhaps "was specimen" should be removed.

Reviewer #3 (Remarks to the Author):

The authors have made improvements, most notably high coverage targeted NGS to support their conclusion that the two tumours that co-existed in this man's prostate and had a differential response to treatment had a different origin and developed independently. This is a bold (and exciting) statement and it's important that this is substantiated. The new data provide complementary supporting data.

The authors have made improvements, most notably high coverage targeted NGS to support their conclusion that the two tumours that co-existed in this man's prostate and had a differential response to treatment had a different origin and developed independently. This is a bold statement and it's important that this is substantiated. The new results provide complementary supporting data.

Suggestions:

1. The rationale for target selection in the custom NGS panel should be included in the main text.

2. The NGS data overlap by tumor region prior to and after treatment should be included graphically in the figure. This is a key result for the authors conclusion that the tumours were independent.

Personally I do not find pictures of IHC helpful and would move to the supplemental data but acknowledge that they pictorially help demonstrate the story. But not at the expense of robust NGS data.

3. I'm unsure what's meant by CLONET had no clonal or subclonal calls. This statement has been added to the methods section and requires clarification.

Response to reviewers' comments.

The authors thank the reviewers for providing another critique of our manuscript in revised form. Where applicable we address each comment below.

Reviewer #1 (Remarks to the Author):

All concerns fully addressed, and the reviewer appreciates the care taken in addressing the challenges of working with limited clinical trial material.

Reviewer #2 (Remarks to the Author):

This is a revised and significantly improved version of the original manuscript reporting a single case of a prostate cancer patient who went through extensive neoadjuvant ADT (goserelin and enzalutamide) prior to RP. The authors have conscientiously addressed my questions. This reviewer has identified only a few minor mistakes/errors in the writing.

1) Supplementary Figure 1 legend: c, should be "...left-sided B1 and right-sided B2/B3 biopsies.....".

Thank you to the eagle-eyed reviewer for spotting this typo.

2) Line 150 beginning: "and" should be "an".

Fixed.

3) Line 356: "... The prostate gland was specimen was serially..." does not make sense and perhaps "was specimen" should be removed.

We removed "gland was" to fix this typo.

Reviewer #3 (Remarks to the Author):

The authors have made improvements, most notably high coverage targeted NGS to support their conclusion that the two tumours that co-existed in this man's prostate and had a differential response to treatment had a different origin and developed independently. This is a bold (and exciting) statement and it's important that this is substantiated. The new data provide complementary supporting data.

Suggestions:

1. The rationale for target selection in the custom NGS panel should be included in the main text.

With the exception of the TP53 mutation, this was random. We decided to verify 50 mutations, as this is the maximum our library design method can accommodate. After selecting all 50, each were inspected in IGV and any coordinate that fell in a region of poor mapping quality in any single sample was excluded. This resulted in the removal of 5 mutations. We then added back the TP53 mutation, which adds up to 46. We have added additional detail to the methods section describing this process.

2. The NGS data overlap by tumor region prior to and after treatment should be included graphically in the figure. This is a key result for the authors conclusion that the tumours were independent.

Personally I do not find pictures of IHC helpful and would move to the supplemental data but acknowledge that they pictorially help demonstrate the story. But not at the expense of robust NGS data.

We have added a new panel, Figure 1j, that depicts two Venn diagrams, each showing the overlap of tumors regions (pre/post treatment) taking into account large somatic copy number alterations and point mutations. Since there is no overlap between the right- and left-sided tumors based on our analysis, these diagrams are separate.

3. I'm unsure what's meant by CLONET had no clonal or subclonal calls. This statement has been added to the methods section and requires clarification.

Our use of CLONET was to determine if analysis of all mutations/SCNAs in a single algorithm would orthogonally identify evidence of clonal or subclonal somatic events that our single-sample analysis had missed. Unlike other samples we have processed with CLONET previously, all mutations from this patient were marked as "uncertain.subclonal" rather than subclonal, or "not.analysed". This is ultimately a negative result. We have clarified this in the text.